# Feline Panleukopenia Virus ZZ202303 Strain: Molecular Characterization and Structural Implications of the *VP2* Gene Phylogenetic Divergence

**DOI:** 10.3390/ijms26104573

**Published:** 2025-05-10

**Authors:** Ming-Yang Wang, Shi-Bo Zhao, Shu-Yi Wang, Meng-Hua Du, Sheng-Li Ming, Lei Zeng

**Affiliations:** 1College of Veterinary Medicine, Henan Agricultural University, Zhengzhou 450046, China; 13346705818@163.com (M.-Y.W.); z1253959849@163.com (S.-B.Z.); 18638626303@163.com (S.-Y.W.); dumenghua2023@163.com (M.-H.D.); 2Key Laboratory of Animal Biochemistry and Nutrition, Ministry of Agriculture and Rural Affairs, Zhengzhou 450046, China; 3Key Laboratory of Animal Growth and Development of Henan Province, Henan Agricultural University, Zhengzhou 450046, China

**Keywords:** FPV, identification of isolates, biological properties, analysis of phylogenetic divergence

## Abstract

Feline panleukopenia virus (FPV), the etiological agent of a highly contagious multispecies disease, demonstrates concerning phylogenetic divergence that compromises vaccine cross-protection. This study aimed to characterize a novel FPV strain through integrated virological and molecular analyses to assess epidemiological implications. From seven clinical specimens obtained from feline hosts with panleukopenia in Henan Province, China, we isolated FPV ZZ202303 using an F81 cell culture coupled with PCR verification, demonstrating potent cytopathic effects (TCID_50_: 10^−5.72^/0.1 mL) and rapid replication kinetics (viral peak at 12–24 h post-infection). Comparative virulence assessments revealed a 1.8- to 2.3-fold greater pathogenicity versus contemporary field strains (2021–2023). Phylogenetic reconstruction based on complete *VP2* gene sequences positioned FPV ZZ202303 within an emerging clade sharing 97.5–98.2% identity with canine parvovirus strains versus 98.8–99.7% with FPV references, forming a distinct cluster (bootstrap = 94%) diverging from vaccine lineages. Critical structural analysis identified a prevalent I101T mutation (89.13% prevalence) in the *VP2* capsid protein’s antigenic determinant region, with molecular modeling predicting altered surface charge distribution potentially affecting host receptor binding. Our findings substantiate FPV ZZ202303 as an evolutionarily divergent strain exhibiting enhanced virulence and unique genetic signatures that may underlie vaccine evasion mechanisms, providing critical data for updating prophylactic strategies against this economically impactful pathogen.

## 1. Introduction

Feline panleukopenia, also known as feline distemper or feline parvovirus infection, is a highly contagious and severe disease caused by FPV. This pathogen primarily affects domestic cats and other felids [1,2]. The disease has no obvious seasonal prevalence, but it peaks in spring and summer. This correlates with feline breeding cycles: concentrated spring births increase the kitten population, while maternal antibodies typically wane at 8–12 weeks, creating susceptible cohorts [3]. Once introduced into a population, FPV can rapidly spread, manifesting as either acute or subacute infections. Regional epidemics are frequently observed, particularly in areas with high feline density [1,4]. Clinical manifestations include high fever, persistent vomiting, severe diarrhea, leukopenia, and, in some cases, ataxia [5]. Notably, FPV is not confined to felids; it also infects a broad range of wild hosts, including weasels and raccoons, highlighting its zoonotic potential [4,6]. Vaccination remains the most effective strategy for FPV prevention. Currently, the primary commercial vaccines available internationally include an inactivated trivalent vaccine developed by Zoetis and a modified live feline triple vaccine produced by Intervet in the Netherlands. However, due to the limited efficacy of imported vaccines against circulating domestic strains, there is an urgent need to accelerate the development of domestic and novel vaccine candidates.

FPV is a member of the genus *Protoparvovirus* within the family Parvoviridae. It is a non-enveloped, single-stranded DNA virus with an icosahedral capsid measuring approximately 20–25 nm in diameter [7,8]. Although FPV is classified as a single serotype, significant variability exists in the pathogenicity and transmissibility of its strains [9]. The genomic length of this virus is approximately 5.2 kb and contains two open reading frames (ORFs). *ORF1* encodes the non-structural proteins *NS1* and *NS2*, which perform multiple biological functions in viral genome replication and assembly, suppression of host innate immunity, and viral pathogenicity [10,11]. *ORF2* encodes the structural proteins *VP1* and *VP2,* with *VP2* constituting 90% of the viral capsid as the major structural component. The VP2 structure comprises eight antiparallel β-strands and five loops: loop 1 (residues 50–100), loop 2 (200–250), loop 3 (300–350), loop 4 (400–450), and the flexible loop 5 (350–400) [12]. The flexible loop determines FPV’s host specificity and hemagglutination activity [13]. Adjacent to this flexible loop in spatial conformation, loop 1 undergoes amino acid variations that alter the flexible loop’s conformation, thereby modulating FPV’s antigenicity, host specificity, and hemagglutination properties [14]. These findings indicate that *VP2* contains critical amino acid residues that determine viral virulence [15], host specificity, and antigenicity, making it a primary target for vaccine and therapeutic development [16,17].

FPV is one of the oldest known viral pathogens in felids and shares high genetic homology with other parvoviruses, including canine parvovirus (CPV), mink enteritis virus (MEV), raccoon parvovirus (RPV), and raccoon dog parvovirus (RDPV), suggesting a common evolutionary origin [18]. FPV exhibits remarkable cross-species transmission capabilities, evolving from FPV to MEV, CPV, and subsequently to the CPV-2a, -2b, and -2c variants, demonstrating exceptional host adaptability [19]. FPV and CPV share 98% nucleotide identity, with antigenic similarity; however, the key amino acid substitutions at positions 80, 93, 103, 323, 564, and 568 alter viral surface topology, receptor-binding specificity, and antigenic epitopes. Phylogenetic evidence indicates that CPV originated from FPV [19]. While CPV-1 initially lacked feline infectivity, later variants (CPV-2a/b/c) have been detected in naturally infected cats, underscoring the ongoing adaptive evolution of FPV [20,21,22]. Consequently, investigating the evolutionary dynamics and driving factors of FPV is essential for understanding its epidemiology and pathogenesis.

According to the *2021 Pet Industry White Paper*, FPV infection has become the leading cause of cat deaths, posing significant risks among feline infectious diseases. In this study, we isolated and characterized a novel FPV strain from the feces of an immunocompetent domestic shorthair cat, analyzed its *VP2* gene and encoded protein, and compared its genetic and antigenic properties with 46 other globally circulating parvovirus strains. The research findings indicate that the *VP2* protein of this strain has typical FPV gene characteristics at the key amino acid sites, with only the I101T (89.13%) site showing a mutation. This mutation at the site leads to infection of immunized cats by FPV, suggesting that this mutation may alter the biological properties of the protein by affecting its spatial conformation. These findings provide critical insights into the functional impact of genomic variations on FPV biology, immunology, and pathogenesis, contributing to the development of more effective preventive and therapeutic strategies.

## 2. Results

### 2.1. PCR-Based Identification of FPV 

Following nucleic acid extraction from the samples, specific primers were designed for four common feline epidemic viruses: FPV, feline calicivirus (FCV), feline coronavirus (FCoV), and feline herpesvirus (FHV). Positive and negative controls were included in the PCR assays. Nucleic acid gel electrophoresis revealed that FCV, FCoV, and FHV were not detected in any of the seven samples (Figure 1A–C). However, specific bands corresponding to the FPV NS1 gene (401 bp) were identified in the samples numbered 4 and 6 (Figure 1D). These results preliminarily confirmed the presence of FPV in these samples and ruled out co-infections with FCV, FCoV, or FHV.

### 2.2. Isolation and Identification of FPV

Building upon the PCR confirmation, FPV nucleic acid-positive samples (Nos. 4 and 6) were inoculated into F81 cells for viral isolation to obtain infectious virions. The F81 cell line, a continuously passaged cell line derived from cat kidneys, is highly susceptible to FPV and can support the efficient replication of the virus. Therefore, it is often used by researchers for the isolation of FPV [20]. Over time, the F81 cells infected with sample 6 exhibited typical cytopathic effects (CPEs), including cell elongation, rounding, and detachment (Figure 2A,B), consistent with the FPV-induced CPEs described in the literature. In contrast, sample 4 showed no CPEs. Consequently, the virus from sample 6 was selected for further study. To confirm viral activity, the isolate was serially passaged five times in F81 cells, with stable CPEs observed after each passage. PCR analysis confirmed the presence of FPV in each passage (Figure 2C), indicating successful viral isolation.

To further validate the isolate, F81 cells were infected with the virus for approximately 36 h, and viral proteins were extracted for Western blot analysis. The *VP2* protein of the isolate was recognized by positive serum (Figure 2D), confirming its identity as FPV. The strain was designated FPV ZZ202303 and purified using plaque purification techniques. The purified virus was subsequently cultured and stored at −80 °C for further analysis.

### 2.3. Indirect Immunofluorescence Assay (IFA) of FPV ZZ202303

To better visualize the cellular infection characteristics of the isolated strain, F81 cells were infected with FPV ZZ202303 for approximately 36 h, fixed, permeabilized, and sequentially incubated with primary and secondary antibodies. Fluorescence microscopy revealed green fluorescence in the infected F81 cells (Figure 3A,B), further confirming the isolate as FPV and designating it as FPV ZZ202303.

### 2.4. TCID_50_ Determination of FPV ZZ202303

TCID_50_ (50% tissue culture infective dose) refers to the viral titer required to induce cytopathic effects in 50% of the inoculated cell cultures, serving as a quantitative measure of viral infectivity. The FPV ZZ202303 virus was serially diluted tenfold in DMEM and used to infect F81 cells. After 5–7 days of observation, the TCID_50_ of the virus was calculated using the Reed–Muench method and determined to be 10^−5.72^/0.1 mL (Figure 4).

### 2.5. Proliferation Kinetics of FPV ZZ202303

Moreover, we conducted viral growth kinetic analysis to delineate replication dynamics. A recombinant PFV-*VP2* plasmid was serially diluted tenfold, and real-time fluorescence quantitative PCR was performed in triplicate. The standard curve exhibited a linear relationship across dilutions ranging from 10^−2^ to 10^−8^ (Figure 5A). F81 cells were infected with FPV ZZ202303, and the supernatants were collected at 12, 24, 36, 48, 60, and 72 h post-infection for viral genomic extraction and quantitative analysis. The results indicated that viral replication peaked between 12 and 24 h, with the highest viral copy number observed at 60 h (Figure 5B).

### 2.6. Nucleotide Sequence Similarity Analysis of the FPV ZZ202303-VP2 Gene

Following characterization of the viral biological properties, we performed multiple sequence alignment and phylogenetic analysis of the *VP2* gene to further elucidate the genetic evolutionary characteristics of FPV ZZ202303. The *VP2* gene of FPV ZZ202303 was compared with domestic and international isolates. The highest nucleotide similarity (98.2%) was observed with CPV strains MW182720.1, FJ005196.1, KC196087.1, and MH329287.1. Similarity to mink enteritis virus (MEV) and raccoon dog parvovirus (RDPV) reference strains ranged from 98.3% to 99.1%. Among the FPV reference strains, similarity ranged from 97.7% to 99.9%, indicating relative conservation of the *VP2* gene during FPV evolution (Figure 6).

### 2.7. Phylogenetic Analysis of the FPV ZZ202303-VP2 Gene

Phylogenetic analysis revealed that FPV ZZ202303 clustered with domestic strains KC473946.1 (Guangdong, China), MG924893.1 (Guangxi, China), MG764510.1 (Sichuan, China), and KX900570.1 (Jilin, China), as well as Thai strain MN127779.1. However, it diverged from foreign strains such as M38246 (Ithaca, NY, USA) and KX434461.1 (Palermo, Italy). Notably, it also formed a distinct branch compared to domestic strains MN908257.1 (Fujian, China) and MH165482.1 (Beijing, China) (Figure 7).

### 2.8. Analysis of Amino Acid Mutation Sites of the FPV ZZ202303-VP2 Protein

After the nucleotide level analysis of the *VP2* gene, we conducted sequence alignment of the amino acids of the entire *VP2* gene using MEGA (https://www.megasoftware.net/, accessed on 2 February 2025) and MegAlign software (https://www.dnastar.com/software/lasergene/megalign-pro/, accessed on 2 February 2025) to further study its protein variation characteristics. The research results indicate that the key residues (80K, 93K, 103V, 323D, 564N, and 568A) were highly conserved and identical to those in the FPV vaccine standard strain CU-4 (Table 1), confirming the isolate as FPV. Comparative analysis revealed one amino acid difference (101F) between FPV ZZ202303 and the CU-4 strain. Differences were also observed with vaccine strains FPV Purevax (101F and 562V) and FPV Felocell (562V). Further comparison with closely related strains KC473946.1 (Guangdong, China) and MN400978.1 (Seoul, Korea) identified multiple divergent amino acid sites in the *VP2* gene. Additionally, the key amino acid differences were observed between FPV ZZ202303 and the CPV reference strain FJ005196.1, highlighting distinct evolutionary pathways.

## 3. Discussion

Infectious diseases represent a significant threat to feline health, with FPV posing a particularly severe risk to domestic and wild felids [23]. FPV was first identified in the first half of the 20th century and successfully isolated in 1957. Since then, the virus has been detected and isolated in multiple regions [23]. Recent studies reveal the expanding host range of FPV, now detected in domestic cats and endangered wildlife, including Siberian tigers, African lions, ferrets, and ring-tailed lemurs [4,24,25]. There is currently no research indicating that FPV can transmit to humans. However, the cross-species transmission of FPV may lead to regional species decline and ecological imbalance, while devastating wildlife-dependent ecotourism, collectively threatening biodiversity conservation, economic stability, and public health. The virus’s environmental resilience further complicates control efforts, as infected and recovered animals can shed the virus, perpetuating transmission [26]. Despite the availability of commercial FPV vaccines, their efficacy is limited by the requirement for multiple doses to induce protective immunity, leaving kittens vulnerable during the immunity gap. More notably, FPV continues to circulate globally with ongoing mutations, resulting in substantial genetic and pathogenic diversity among regional strains [14,27]. Current vaccines demonstrate limited efficacy against certain variants, as evidenced by FPV-positive cases detected even in fully vaccinated cats, indicating poor cross-protection against emerging strains [4].

To address these challenges, this study employed a synchronized infection technique to optimize FPV isolation and amplification, enhancing viral replication and purification efficiency [24]. From seven fecal and oral/nasal secretion samples collected at two pet hospitals in Henan Province, a novel FPV strain, designated FPV ZZ202303, was successfully isolated (Figure 1, Figure 2 and Figure 3). The isolate exhibited a TCID_50_ of 10^−5.72^/0.1 mL (Figure 4), a relatively high viral titer compared to 41 FPV strains isolated nationwide between 2019 and 2024, which ranged from 10^−4.33^ to 10^−6.33^ [28]. One-step growth curve analysis revealed that viral replication peaked between 12 and 24 h post-infection, with the highest viral copy number observed at 60 h (Figure 5).

VP2, the primary capsid protein of FPV, plays a critical role in viral recognition, host cell receptor binding, and cellular invasion. Nucleotide similarity analysis of the VP2 gene from FPV ZZ202303 showed a >97.7% identity with reference parvovirus sequences, up to 99.7%, indicating high genetic conservation (Figure 6). Phylogenetic analysis based on VP2 divided the evolutionary tree into three clades, with FPV ZZ202303 clustering closely with such domestic strains as MG924893.1 (Guangxi, China), MG764510.1 (Sichuan, China), and KP280068.1 (Heilongjiang, China), as well as international strains MN400978.1 (Seoul, Korea) and MN127779.1 (Bangkok, Thailand). This suggests recent evolutionary divergence, potentially driven by human-mediated viral spread. Notably, FPV ZZ202303 shared a clade with vaccine strain M38246.1 (CU-4) but diverged from EU498681.1 (Felocell) and EU498680.1 (Purevax), raising questions about the efficacy of the existing vaccines against locally circulating strains (Figure 7).

Comparative analysis of *VP2* amino acid sequences from 46 parvovirus strains revealed 1–2 amino acid differences in FPV ZZ202303, with high mutation frequencies observed at positions A91S (28.26%) and I101T (89.13%) (Table 2). Among the 13 A91S mutant strains, 69.23% (9/13) were identified as locally circulating Henan variants, while FPV ZZ202303 retained the prototype residue at this position, suggesting potential introduction through interregional transmission. Notably, the mutation at position 101 in the VP2 protein of FPV ZZ202303 exhibits high concordance with the currently prevalent strains. Although I101T lies outside the *VP2* Loop1 domain (residues 87–93 aa), it localizes to the receptor-binding core and an antibody epitope region [29,30]. Structural analyses revealed that the I101T substitution induces conformational changes in the receptor-binding interface via polar interactions with Asp99 [31]. This mutation has been detected in canine isolates, implying cross-species transmission potential, though extensive experimental validation remains necessary. Of particular concern, the I101T mutation likely compromises neutralizing antibody efficacy by altering epitope topology, consistent with our isolation of this strain from a fully vaccinated cat. These findings indicate that current vaccines inadequately address emerging variants carrying this critical mutation.

Recent studies have shown that while the FPV-*VP2* protein is relatively conserved during evolution, significant biological mutations have begun to appear. For example, the Giant Panda/CD/2018 strain isolated from the giant panda was the first to show a VP2 G299E mutation. This mutation at the site may alter the protein’s spatial conformation, thereby expanding the virus’s host range [32]. In the feline FPV-251 strain, a *VP2* A300P amino acid substitution was detected, and this variant demonstrated effective replication in canine cell lines, suggesting it may have gained the potential to infect canids [33]. These findings indicate that despite the overall conservation of the *VP2* protein, mutations at the key sites could still lead to an expansion of the virus’s host range, potentially reducing the efficacy of vaccines.

In addition to the antigenic drift of FPV leading to vaccine efficacy attenuation, maternal antibodies can also interfere with vaccine-mediated protection. Studies have shown that FPV infection in kittens follows a herd immunity principle. When a sufficient proportion of adult cats in a population are vaccinated, environmental viral transmission pressure decreases, thereby reducing infection risk in kittens. However, high levels of maternal antibodies in kittens may neutralize attenuated vaccines, resulting in vaccination failure [34]. This phenomenon has prompted the development of novel vaccine strategies. On the one hand, mRNA vaccines or adenovirus vector platforms can be engineered to dynamically incorporate critical antigenic information from prevalent *VP2* protein mutations (e.g., I101T). These platforms enable dynamic integration of critical antigenic information from prevalent *VP2* protein mutations (e.g., I101T). These next-generation candidates enable real-time antigenic matching with circulating variants, thereby enhancing protective efficacy against evolving viral strains. A study used attenuated feline herpesvirus type 1 (FHV-1) as a vector, inserting the *VP2* antigen gene of FPV. This recombinant live vector vaccine provides good immune protection against both FPV and FHV and possesses the characteristics of a multivalent vaccine [35]. On the other hand, *VP2*-based virus-like particle (VLP) vaccines have been developed. Experimental studies demonstrate that VLP vaccines produced via prokaryotic (*E. coli*) or eukaryotic (insect cell-baculovirus) expression systems can induce high-titer neutralizing antibodies in animal models [36,37].

In this study, we successfully isolated and characterized an FPV strain from Henan Province, China. Phylogenetic analysis revealed high conservation of the *VP2* gene, suggesting a potential non-local evolutionary origin. However, the limited scope of a single strain (*n* = 1) from one geographic region (Henan) restricts a comprehensive understanding of FPV’s broader evolutionary dynamics. Furthermore, while high-frequency mutations at the key *VP2* residues (e.g., I101T) were identified, their functional consequences on viral properties remain experimentally unvalidated, and the protective efficacy of current vaccines against prevalent strains was not assessed. Given the ongoing challenges posed by FPV, it is crucial to conduct systematic epidemiological surveys and genomic monitoring. We can establish a global FPV genome database, perform phylogenetic analysis, and track mutations; secondly, we should enhance host range investigations to scientifically assess the risk of cross-species transmission. In terms of control strategies, it is essential to implement vaccine prevention measures effectively, increasing vaccination coverage to build herd immunity and reduce viral transmission risks [38]. At the same time, we should utilize diverse detection technologies to accurately identify infections at an early stage, providing a scientific basis for timely control measures. Ultimately, an integrated systemic network combining vaccination, early warning systems, viral surveillance, and scientific research should be established. This comprehensive approach will provide robust protection against FPV epidemics.

## 4. Materials and Methods

### 4.1. Sample Source

Seven clinical samples (including feces and ocular/nasal secretions) were collected from suspected feline panleukopenia cases at Kangxu pet hospitals and Luoyang Beikangmei Pet Hospital. The specific information of the samples is shown in Table 3. FPV-positive samples were isolated and preserved by the Key Laboratory of Animal Chemistry and Nutrition of the Ministry of Agriculture and Rural Areas.

### 4.2. Primer Synthesis

Based on the standard strain sequences available at the NCBI, four specific primers for identifying cat viruses were designed: FPV-NS1, FHV-gB, FCV-VP1, and FIPV-N, along with FPV-VP2 identification primers, FPV full-length VP2 amplification primers, and FPV real-time fluorescence quantitative PCR amplification primers. The primers were synthesized by Qingke Biotechnology Co., Ltd., Beijing, China. The primers are shown in Table 4.

### 4.3. Primary Reagents

The FastPure Viral DNA/RNA Mini Kit was purchased from Nanjing Novogene Bioinformatics Technology Co., Ltd., Nanjing, China; the All-in-one First-strand Synthesis MasterMix was obtained from Kemix Biotechnology Co., Ltd., Zhengzhou, China; the Prime STAR Max DNA Polymerase Kit was purchased from Beijing Baorui Medicine Biotechnology Co., Ltd., Beijing, China; and the one-step ZTOPO-Blunt/TA Kit was purchased from Beijing Zhuangmeng International Bio-Gene Technology Co., Ltd., Beijing, China.

### 4.4. Sample Treatment

About 1 mL of PBS was added to each collected disease material and then vortexed. The samples were then incubated overnight at 4 °C to facilitate virus release. Subsequently, they were subjected to centrifugation at 8000× *g* for 5 min. The supernatant was carefully decanted and passed through a 0.22 μm filter to eliminate most bacterial contaminants. The filtered supernatant was finally stored at −80 °C for subsequent analysis and use.

### 4.5. PCR Identification of FPV

Viral nucleic acids were isolated from the collected samples following the protocol provided with the FastPure Viral DNA/RNA Mini Kit nucleic acid extraction kit. Then, reverse transcription was performed according to the instructions for the All-in-one First-strand Synthesis MasterMix kit to convert the extracted RNA into complementary DNA (cDNA). Specific primers were designed for the identification of FPV, feline calicivirus (FCV), feline coronavirus (FCoV), and feline herpesvirus (FHV), and target gene fragments were amplified using polymerase chain reaction (PCR). PCR was performed in a 10 μL system containing 5 μL of 2× Rapid Taq Master Mix, 1 μM each of the forward (F) and reverse (R) primers, and 0.1–1 μg of the FPV genomic DNA template, with the remaining volume supplemented by nuclease-free ddH_2_O. The amplification protocol included an initial denaturation at 95 °C for 3 min, followed by 35 cycles of denaturation (95 °C, 15 s), annealing (56 °C, 15 s), and extension (72 °C, 10 s), with the final extension at 72 °C for 5 min using a thermal cycler. Amplified products were subsequently subjected to 2% agarose gel electrophoresis for validation of target fragment amplification.

### 4.6. Isolation and Cultivation of FPV

After PCR confirmation of FPV positivity, the samples were inoculated onto F81 cells at a cell density of roughly 60% confluence. The infected cultures were then incubated at 37 °C under a 5% CO_2_ environment to facilitate viral replication, with close observation for the development of cytopathic effects. In cases where cytopathic effects did not manifest, the culture supernatant was collected and subjected to a freeze–thaw cycle at −80 °C to release virus particles, before being used to inoculate fresh F81 cells for additional passages. The virus-containing supernatant from each passage was reanalyzed by PCR to verify the continued presence of FPV. If cytopathic effects were still absent by the fifth passage, the sample was deemed unsuitable and discarded. However, virus isolation was confirmed in samples demonstrating sustained cytolytic plaques (manifested as cell elongation and network formation, cytoplasmic rounding, and detachment) through five consecutive passages, with PCR-confirmed FPV positivity in all passage supernatants. Subsequently, the isolated viral strains were subjected to plaque purification on F81 cell monolayers. Viral harvest and cryopreservation were initiated upon observation of 80% CPE.

### 4.7. Immunoblotting Analysis of FPV

The F81 cells infected with the FPV isolate were cultured until cytopathic effects were evident, but prior to cell detachment. Cell samples were harvested, and a negative control was set up. Fifty microliters of cell lysis buffer were added to the cell samples, which were repeatedly drawn with a syringe until the samples were clear. The lysed samples were then centrifuged at 12,000× *g* for 5 min at 4 °C to separate the supernatant, which was used for the BCA protein assay to determine protein concentrations. The viral protein samples were analyzed using SDS-PAGE and subsequently transferred to a nitrocellulose membrane. The membrane was blocked with 5% non-fat dry milk at room temperature for 1 h to prevent non-specific binding. After washing with TBST, it was incubated overnight at 4 °C with an FPV-positive serum. Following another TBST wash, the membrane was incubated with a horseradish peroxidase (HRP)-conjugated secondary antibody specific to cat immunoglobulins (diluted 1:5000) at room temperature for 1 h, and the results were photographed for observation.

### 4.8. IFA Identification of FPV

F81 cells were digested and inoculated into a 24-well plate. Once the cell density reached ca. 60%, the FPV isolate was inoculated into the cells, which were then placed in a 37 °C incubator with 5% CO_2_ for cultivation. Ca. 36 h after infection, the culture medium was discarded, and 200 μL of 4% paraformaldehyde was added to each well for fixation at room temperature for 30 min. The wells were washed three times with PBS, and 200 μL of 0.1% Triton X-100 was added to each well for permeabilization at room temperature for 5 min. After washing with PBS, 200 μL of 10% FBS in PBS was added to each well for blocking at room temperature for 1 h. Following another PBS wash, FPV-positive serum was added and incubated overnight at 4 °C. After three washes with PBS, a green fluorescent-labeled secondary antibody against cats (1:1000) was added and incubated at room temperature for 1 h. After PBS washing, fluorescence was observed under a fluorescence microscope (Zeiss LSM 800, Jena, Germany).

### 4.9. TCID_50_ Determination of FPV

F81 cells were digested and inoculated at a density of 1 × 10^5^ cells per well in a 96-well plate. Once the cells grew to ca. 40%, the virus was diluted tenfold for inoculation, ranging from 10^−1^ to 10^−10^. A blank F81 cell control group was also set up. After 1 h of adsorption in a 37 °C incubator with 5% CO_2_, the maintenance medium was replaced, and the cells were incubated for another 5–7 days, observing and recording the number of wells showing cytopathic effects. The TCID_50_ of the FPV isolate was calculated according to the Reed–Muench method.

### 4.10. Proliferation Kinetics Analysis of FPV

#### 4.10.1. Establishing a Standard Curve

The FPV-VP2 recombinant standard plasmid stored in the Key Laboratory of Animal Chemistry and Nutrition, Ministry of Agriculture and Rural Affairs, was quantified for concentration and subjected to tenfold serial dilutions. Each sample was analyzed in triplicate using quantitative real-time PCR (qPCR). A standard curve for FPV-VP2 was generated by plotting the mean Ct values (y-axis) against the logarithm of template copy numbers (x-axis).

#### 4.10.2. Collecting Virus Liquid Samples at Different Time Points

The FPV isolate was diluted tenfold and inoculated into F81 cells cultured on 35 mm plates. After 1 h of adsorption in the cell incubator, PBS was used for washing, and a 1% cell maintenance solution was added for continued cultivation. Virus liquid samples were collected at 0 h, 12 h, 24 h, 36 h, 48 h, 60 h, and 72 h, and viral nucleic acids were extracted for real-time fluorescence quantification.

#### 4.10.3. Phylogenetic Divergence of FPV

Following the extraction of nucleic acids from the viral suspension, a set of specific primers was designed, targeting the VP2 gene sequence of the FPV CU-4 standard strain, as referenced in the GenBank database (accession number M38246.1). These primers were utilized to amplify the complete VP2 gene of FPV via PCR. The PCR products were then separated by electrophoresis on a 1% agarose gel. Upon successful identification of the target bands, they were excised from the gel and purified. Using the one-step ZTOPO-Blunt/TA cloning kit(Zoman Biotechnology, Beijing, China), the purified target fragments were ligated into the TA vector for subsequent prokaryotic expression, followed by the selection of single colonies for Sanger sequencing analysis.

To analyze mutation profiles and evolutionary relationships of VP2 amino acids in the isolated strains, multiple sequence alignment was performed using the ClustalW algorithm in MEGA 7.0 (https://www.megasoftware.net/, accessed on 2 February 2025), aligning target sequences with the VP2 reference strains retrieved from GenBank. Phylogenetic tree construction was subsequently conducted based on the neighbor-joining method with 1000 bootstrap replicates, employing the Poisson correction model for evolutionary distance calculation. Tree topology was visualized and refined using EvolView v3 (https://www.evolgenius.info/evolview, accessed on 2 February 2025). Finally, nucleotide homology analysis was conducted using MegAlign’s View-Alignment Report to generate a sequence identity matrix (https://www.dnastar.com/software/lasergene/, accessed on 2 February 2025).

### 4.11. Statistical Analysis

All the data are displayed as the means ± standard deviations (SDs) and were analyzed via Prism 8.3 (GraphPad). One-way ANOVA tests, or *t*-tests, were performed for statistical analysis (*p* < 0.05 was considered to indicate a significant difference).

### 4.12. Statement of Informed Consent

The owners of the animals agreed in writing that the animals’ test samples were used in the study.

## Figures and Tables

**Figure 1 ijms-26-04573-f001:**
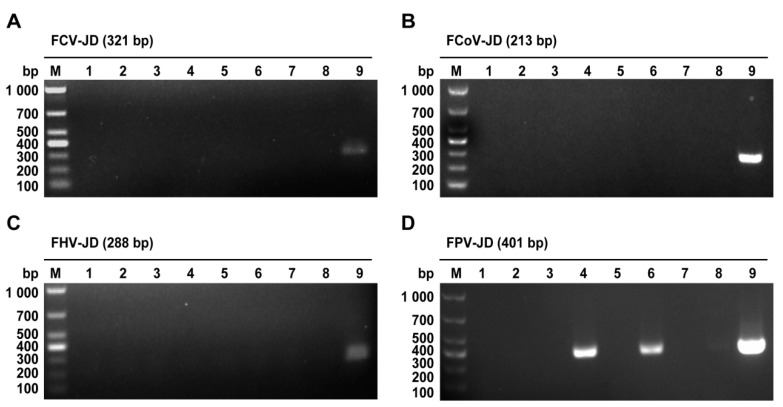
PCR detection of the pathological samples. (**A**) FCV identification of the disease samples; (**B**) FCoV identification of the disease samples; (**C**) FHV identification of the disease samples; (**D**) FPV identification of the disease samples. Note: M: DL1,000 DNA marker; 1–7: sample to be tested; 8: the template was a negative control of sterile water; 9: positive control.

**Figure 2 ijms-26-04573-f002:**
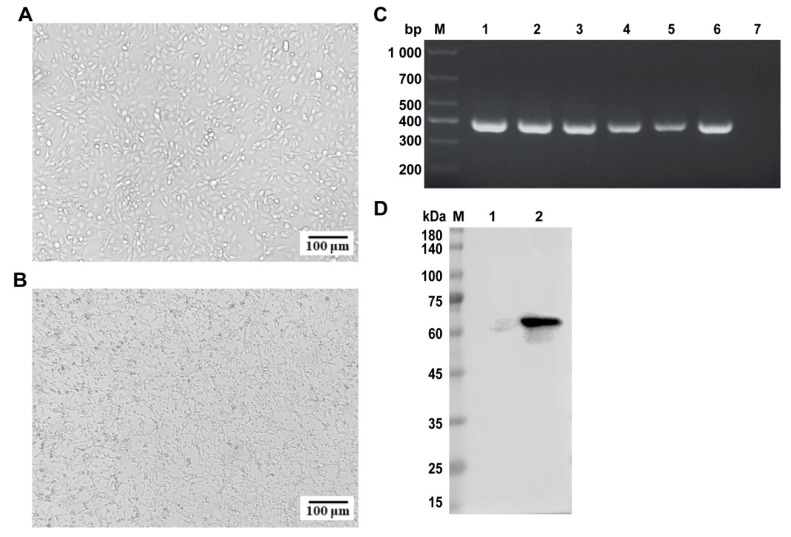
Isolation and identification of FPV. (**A**) Normal F81 cell morphology; (**B**) F81 cell morphology after FPV infection; (**C**) PCR detection of the samples after continuous passage; (**D**) Westen blotting identification of FPV. Bar: 100 μm. M: DL1000 DNA marker; 1–5: five successive generations of the disease venom; 6: positive control; 7: the template was a negative control of sterile water.

**Figure 3 ijms-26-04573-f003:**
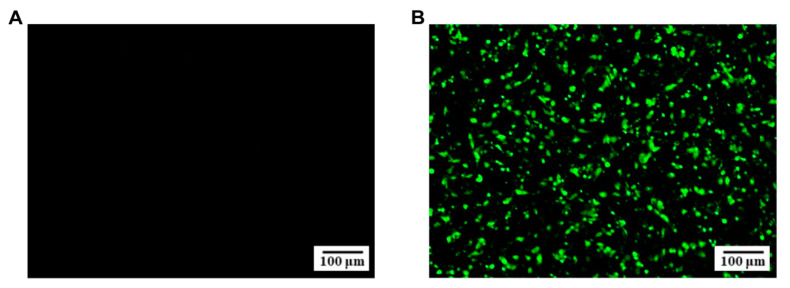
IFA identification of FPV ZZ202303. (**A**) Normal F81 cell; (**B**) F81 cell after FPV infection, the green fluorescence observed in the cells indicates infection with FPV. Bar: 100 μm.

**Figure 4 ijms-26-04573-f004:**
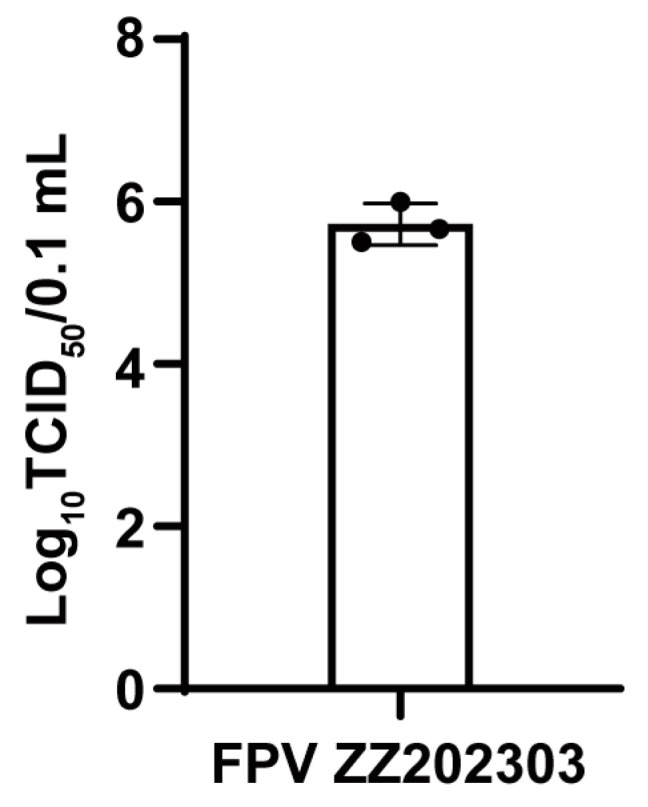
Determination of TCID_50_ of FPV ZZ202303.

**Figure 5 ijms-26-04573-f005:**
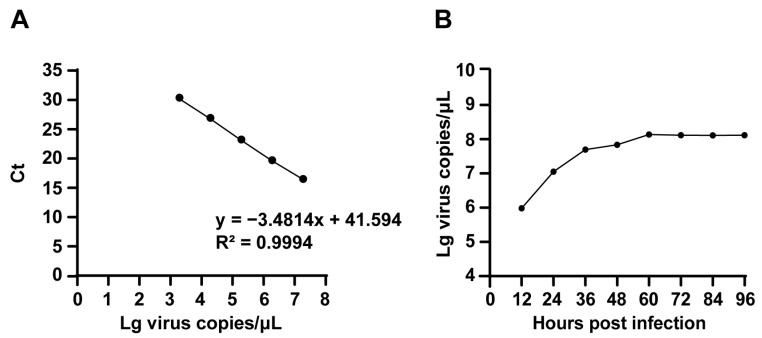
Proliferation kinetics analysis of FPV ZZ202303. (**A**) Standard curve of the FPV-*VP2* recombinant plasmid; (**B**) one-step growth curve of FPV ZZ202303.

**Figure 6 ijms-26-04573-f006:**
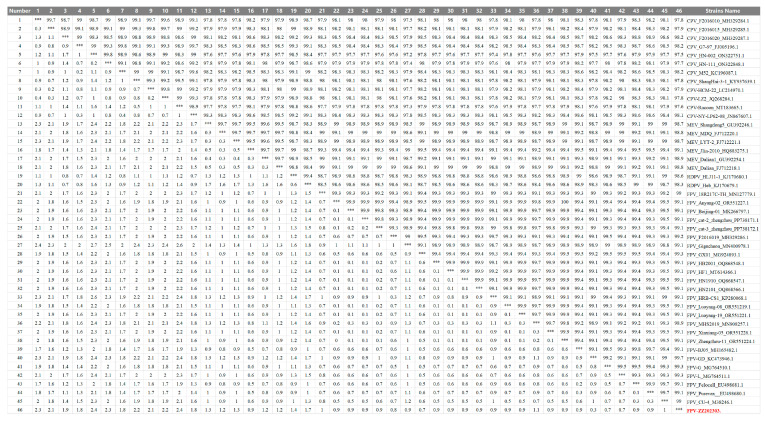
Nucleotide Sequence similarity analysis of the FPV ZZ202303-*VP2* gene. Analysis of nucleotide similarity of the VP2 gene of FPV ZZ202303. Note: FPV ZZ202303 is the isolate from this study. *** This is the diagonal, with the upper part and the lower part separated.

**Figure 7 ijms-26-04573-f007:**
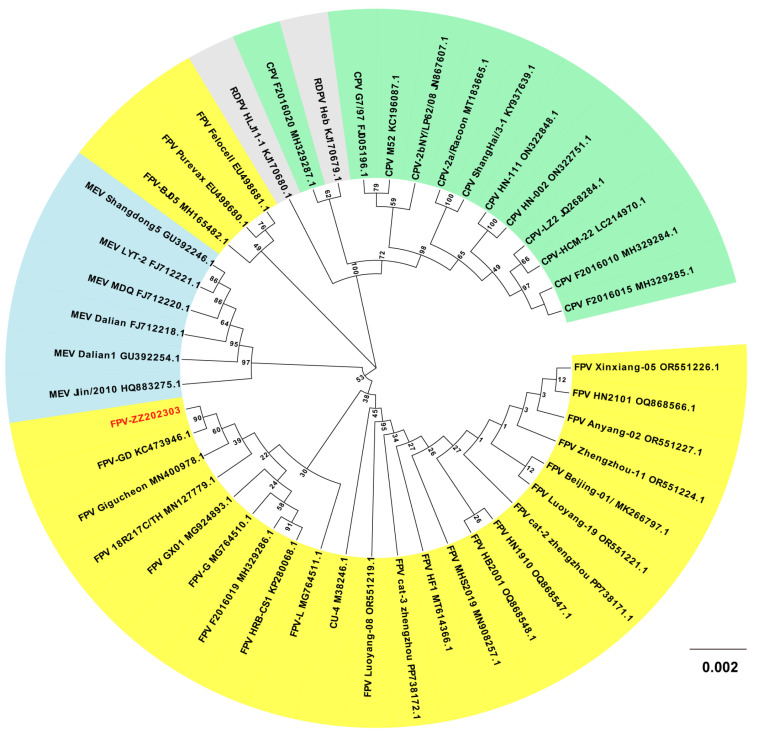
Phylogenetic analysis of the FPV ZZ202303-*VP2* gene. FPV ZZ202303 phylogenetic divergence tree of the VP2 genes. Note: FPV ZZ202303 is the isolate from this study. Different colors represent different hosts of parvovirus. Yellow, blue, green, and gray represent FPV, MEV, FCV, and RDPV, respectively. The values on the branches of the phylogenetic tree are bootstrap values. The higher the value, the higher the credibility of the corresponding branch.

**Table 1 ijms-26-04573-t001:** FPV-VP2 reference sequences.

Number	Strain	GenBank A ccession No.	Isolation Location	Isolation Time
**1**	F2016010	MH329284.1	Luoyang, China	2016
**2**	F2016015	MH329285.1	Luoyang, China	2016
**3**	F2016020	MH329287.1	Luoyang, China	2016
**4**	G7/97	FJ005196.1	Bari, Italy	1997
**5**	HN-002	ON322751.1	Zhengzhou, China	2020
**6**	HN-111	ON322848.1	Zhengzhou, China	2020
**7**	M52	KC196087.1	Montevideo, Uruguay	2006
**8**	ShangHai/3-1/2016	KY937639.1	ShangHai, China	2016
**9**	CPV/dog/HCM/22/2013	LC214970.1	Tokyo, Japan	2013
**10**	CPV-LZ2	JQ268284.1	Lanzhou, China	2011
**11**	CPV-2a/Racoon_dog/QHD/2/19	MT183665.1	Jilin, China	2019
**12**	CPV-2b/Dog/NY/LP62/08	JN867607.1	Ithaca, USA	2008
**13**	Shangdong5	GU392246.1	Shangdong, China	2007
**14**	MDQ	FJ712220.1	Jilin, China	2008
**15**	LYT-2	FJ712221.1	Jilin, China	2008
**16**	Jlin/2010	HQ883275.1	Jilin, China	2010
**17**	Dalian1	GU392254.1	Dalian, China	2009
**18**	Dalian	FJ712218.1	Dalian, China	2008
**19**	RDPV HLJ11-1	KJ170680.1	Jilin, China	2011
**20**	RDPV Heb10-2	KJ170679.1	Baerbin, China	2010
**21**	18R217C/TH/2018	MN127779.1	Thailand	2018
**22**	Anyang-02	OR551227.1	Anyang, China	2021
**23**	Beijing-01/2018	MK266797.1	Beijing, China	2017
**24**	cat-2	PP738171.1	Zhengzhou, China	2021
**25**	cat-3	PP738172.1	Zhengzhou, China	2021
**26**	F2016019	MH329286.1	Zhengzhou, China	2019
**27**	Gigucheon	MN400978.1	Seoul, Korea	2017
**28**	GX01	MG924893.1	Guilin, China	2016
**29**	HB2001	OQ868548.1	Baoding, China	2020
**30**	HF1	MT614366.1	Hefei, China	2019
**31**	HN1910	OQ868547.1	Zhengzhou, China	2019
**32**	HN2101	OQ868566.1	Zhengzhou, China	2021
**33**	HRB-CS1	KP280068.1	Harbin, China	2014
**34**	Luoyang-08	OR551219.1	Luoyan, China	2020
**35**	Luoyang-19	OR551221.1	Luoyan, China	2020
**36**	MHS2019	MN908257.1	Longyan, China	2019
**37**	Xinxiang-05	OR551226.1	Xinxiang, China	2022
**38**	Zhengzhou-11	OR551224.1	Zhengzhou, China	2021
**39**	FPV-BJ05	MH165482.1	Beijing, China	2014
**40**	FPV-GD(12/09/YGP)	KC473946.1	Guangdong, China	2012
**41**	FPV-G	MG764510.1	Chengdu, China	1999
**42**	FPV-L	MG764511.1	Chengdu, China	2015
**43**	Felocell	EU498681.1	Bari, Italy	2008
**44**	Purevax	EU498680.1	Bari, Italy	2008
**45**	CU-4	M38246.1	Ithaca, USA	1996

**Table 2 ijms-26-04573-t002:** Amino acid mutation site of the FPV ZZ202303-VP2 protein.

Strain	Amino Acid Mutation Site												
	9	16	19	80	93	101	103	305	323	350	562	564	568
FPV ZZ202303	D	R	R	K	K	F	V	D	D	Q	V	N	A
CU-4 (M38246.1)	D	R	R	K	K	I	V	D	D	Q	V	N	A
Felocell (EU498681.1)	D	R	R	K	K	F	V	D	D	Q	L	N	A
Purevax (EU498680.1)	D	R	R	K	K	I	V	D	D	Q	L	N	A
Gigucheon (MN400978.1)	N	K	K	K	K	F	V	D	D	Q	V	N	A
FPV-GD (KC473946.1)	D	R	R	K	K	F	V	D	D	R	V	N	A
G7/97 (FJ005196.1)	D	R	R	R	N	F	A	Y	N	Q	V	S	G

Note: red letters indicate the amino acids that mutated after the FPV strain.

**Table 3 ijms-26-04573-t003:** Sample information.

Case Number	Gender	Months of Age	Breed	Vaccination Status	Immunity Status
1	Female	2	British Shorthair	No	Unimmunized
2	Female	7	Ragdoll	Yes	Complete immunity
3	Female	12	Chinese civet cat	Yes	Complete immunity
4	Male	9	American Shorthair	Yes	Complete immunity
5	Female	8	British Shorthair	Yes	Complete immunity
6	Male	6	Chinese civet cat	Yes	Complete immunity
7	Male	21	Maine Coon	Yes	Complete immunity

**Table 4 ijms-26-04573-t004:** Primer sequences.

Primer Name	Primer Sequence 5′-3′	Fragment Size, bp
FPV-NS1-F	TGTACTTTGCGGGACTTGGT	401
FPV-NS1-R	ACATTACCCACAGCTTGTGCT	
FHV-gB-F	GACGTGGTGAATTATCAGC	288
FHV-gB-R	CAACTAGATTTCCACCAGGA	
FCV-VP1-F	CAARGGAGAAAATTCDGACGA	321
FCV-VP1-R	GTATTTWAGCACGTTAGCGCAGGT	
FIPV-N-F	GGCAACCCGATGTTTAAAACTGG	214
FIPV-N-R	CACTAGATCCAGACGTTAGCTC	
FPV-VP2-JDF	GGAACTAGTGGCACACCAACA	386
FPV-VP2-JDR	GGCCCTTGTGTAGACGCTT	
FPV-VP2-F	GTATACCATATAACAAACCTTC	1903
FPV-VP2-R	ATGAGTGATGGAGCAGTTCAAC	
Q-FPV-VP2-F	GTATACCATATAACAAACCTTC	130
Q-FPV-VP2-R	TCATAGCTGCTGGAGTAAATGG	

## Data Availability

The original contributions presented in this study are included in the article. Further inquiries can be directed to the corresponding authors.

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
