# Peer review of "Feline Panleukopenia Virus ZZ202303 Strain: Molecular Characterization and Structural Implications of the VP2 Gene Phylogenetic Divergence"

_ijms, 2025, doi:10.3390/ijms26104573_

Round 1

Reviewer 1 Report

Comments and Suggestions for Authors

The manuscript titled "Isolation and Identification of Feline Panleukopenia Virus ZZ202303 Strain and Genetic Evolution Analysis of VP2 Gene" offers a valuable contribution to veterinary virology by isolating and characterizing a novel FPV strain, ZZ202303, and analyzing its genetic evolution, particularly focusing on mutations in the VP2 protein. The work is original and fills a gap in the understanding of FPV's genetic diversity, offering insights into its evolution and implications for vaccine development. To enhance the manuscript, I recommend revising the language for clarity, expanding the discussion on vaccine implications, and ensuring that the novelty of the findings is more explicitly highlighted. Overall, this manuscript presents valuable data that will aid in the future control and prevention of FPV.

Major:

(1) The manuscript mentions an important mutation in the VP2 gene (I101T), but the functional impact of this mutation on the virus's biology and pathogenesis is not explored. A more detailed investigation of how this mutation influences viral infectivity, immune evasion, or vaccine efficacy would be beneficial.

(2) Although the manuscript notes that the mutation may impact vaccine efficacy, the discussion lacks a comprehensive evaluation of how this new strain might affect existing vaccine strategies. A clearer exploration of potential vaccine adaptation strategies would make the work more impactful.

(3) The study primarily relies on a single FPV strain (ZZ202303), making it difficult to generalize the findings to other FPV strains. The experimental analysis would be more robust with comparisons to a broader range of FPV strains from different geographical regions or years.

(4) The methods section provides sufficient detail on the isolation and identification of the FPV strain. However, the authors could provide more information about the specific techniques used for the phylogenetic analysis, including the software version and settings used.

(5) Figure 4 shows a bar graph for TCID50 but labels the isolate as "FPV ZZ202306" rather than "FPV ZZ202303."

Minor:

(1) Figures and tables are useful, but some figure legends would benefit from more context for clarity. While the manuscript's length is appropriate, there are minor issues with grammar and phrasing that should be refined to improve overall clarity and flow.

(2) There are occasional grammatical errors and awkward phrasing, which make some sections difficult to follow. For example, the sentence "Comparative analysis revealed that FPV ZZ202303 demonstrated heightened virulence relative to other FPV strains isolated in the preceding two years" could be more concise: "Comparative analysis showed that FPV ZZ202303 exhibited higher virulence than other FPV strains isolated in the last two years."

(3) The paper cites relevant studies in the field, but there could be more references to recent research on FPV and related viruses, particularly studies that address the implications of genetic mutations in viral vaccines. The references should be updated to include the latest literature on FPV evolution and vaccination.

(4) The manuscript mentions FPV's zoonotic potential but does not thoroughly discuss implications for wildlife conservation or public health.

Author Response

Review1

The manuscript titled "Isolation and Identification of Feline Panleukopenia Virus ZZ202303 Strain and Genetic Evolution Analysis of VP2 Gene" offers a valuable contribution to veterinary virology by isolating and characterizing a novel FPV strain, ZZ202303, and analyzing its genetic evolution, particularly focusing on mutations in the VP2 protein. The work is original and fills a gap in the understanding of FPV's genetic diversity, offering insights into its evolution and implications for vaccine development. To enhance the manuscript, I recommend revising the language for clarity, expanding the discussion on vaccine implications, and ensuring that the novelty of the findings is more explicitly highlighted. Overall, this manuscript presents valuable data that will aid in the future control and prevention of FPV.

Major

(1) The manuscript mentions an important mutation in the VP2 gene (I101T), but the functional impact of this mutation on the virus's biology and pathogenesis is not explored. A more detailed investigation of how this mutation influences viral infectivity, immune evasion, or vaccine efficacy would be beneficial.

Response: We sincerely appreciate the reviewers' valuable feedback. In response to your suggestions, we have incorporated an in-depth discussion into the Discussion section regarding the potential impacts of this mutation on viral infectivity, immune evasion, and vaccine efficacy. This addition is expected to provide further insights into the biological significance of the mutation. The detailed content is as follows: Notably, the mutation at position 101 in the VP2 protein of FPV ZZ202303 exhibits high concordance with currently prevalent strains. Although I101T lies outside the VP2 Loop1 domain (residues 87–93 aa), it localizes to the receptor-binding core and an antibody epitope region. Structural analyses revealed that the I101T substitution induces conformational changes in the receptor-binding interface via polar interactions with Asp99. This mutation has been detected in canine isolates, implying cross-species transmission potential, though extensive experimental validation remains necessary. Of particular concern, the I101T mutation likely compromises neutralizing antibody efficacy by altering epitope topology, consistent with our isolation of this strain from a fully vaccinated cat. These findings indicate that current vaccines inadequately address emerging variants carrying this critical mutation.

(2) Although the manuscript notes that the mutation may impact vaccine efficacy, the discussion lacks a comprehensive evaluation of how this new strain might affect existing vaccine strategies. A clearer exploration of potential vaccine adaptation strategies would make the work more impactful.

Response: We sincerely appreciate the constructive suggestions provided by the reviewers. In response to your feedback, we have conducted a detailed analysis of potential vaccine adaptation strategies in the Discussion section. The revised content is presented as follows: Of particular concern, the I101T mutation likely compromises neutralizing antibody efficacy by altering epitope topology, consistent with our isolation of this strain from a fully vaccinated cat. These findings indicate that current vaccines inadequately address emerging variants carrying this critical mutation.In addition to antigenic drift of FPV leading to vaccine efficacy attenuation, maternal antibodies can also interfere with vaccine-mediated protection. Studies have shown that feline panleukopenia virus (FPV) infection in kittens follows a herd immunity principle. When a sufficient proportion of adult cats in a population are vaccinated, environmental viral transmission pressure decreases, thereby reducing infection risk in kittens. However, high levels of maternal antibodies in kittens may neutralize attenuated vaccines, resulting in vaccination failure. This phenomenon has prompted the development of novel vaccine strategies. On one hand, mRNA vaccines or adenovirus vector platforms can be engineered to dynamically incorporate critical antigenic information from prevalent VP2 protein mutations (e.g., I101T). These platforms enable dynamic integration of critical antigenic information from prevalent VP2 protein mutations (e.g., I101T). These next-generation candidates enable real-time antigenic matching with circulating variants, thereby enhancing protective efficacy against evolving viral strains. On the other hand, VP2-based virus-like particle (VLP) vaccines have been developed. Experimental studies demonstrate that VLP vaccines produced via prokaryotic (E. coli) or eukaryotic (insect cell-baculovirus) expression systems can induce high-titer neutralizing antibodies in animal models.

(3) The study primarily relies on a single FPV strain (ZZ202303), making it difficult to generalize the findings to other FPV strains. The experimental analysis would be more robust with comparisons to a broader range of FPV strains from different geographical regions or years.

Response: We sincerely appreciate the reviewers' valuable suggestions. This study primarily focuses on a single FPV strain (ZZ202303). In future research, it would be advantageous to conduct comparisons with a wider range of FPV strains originating from diverse geographical regions or collected in different years. Nevertheless, the current study represents only preliminary findings. We are highly grateful for the reviewers' insightful feedback and will further advance our research and analysis in line with their recommendations.

(4) The methods section provides sufficient detail on the isolation and identification of the FPV strain. However, the authors could provide more information about the specific techniques used for the phylogenetic analysis, including the software version and settings used.

Response: To analyze mutation profiles and evolutionary relationships of VP2 amino acids in isolated strains, multiple sequence alignment was performed using the ClustalW algorithm in MEGA 7.0 (https://www.megasoftware.net/), aligning target sequences with VP2 reference strains retrieved from GenBank. Phylogenetic tree construction was subsequently conducted based on the Neighbor-Joining method with 1000 bootstrap replicates, employing the Poisson correction model for evolutionary distance calculation. Tree topology was visualized and refined using EvolView (https://www.evolgenius.info/evolview). Finally, nucleotide homology analysis was implemented through MegAlign's View-Alignment Report to generate a sequence identity matrix (https://www.dnastar.com/software/lasergene/).

(5) Figure 4 shows a bar graph for TCID50 but labels the isolate as "FPV ZZ202306" rather than "FPV ZZ202303."

Response: Firstly, we extend our sincerest apologies to the reviewers for our oversight. Secondly, we are deeply appreciative of the reviewers' meticulous attention in identifying the errors. The content of the image has now been corrected accordingly.

Figure 4

Minor

(1) Figures and tables are useful, but some figure legends would benefit from more context for clarity. While the manuscript's length is appropriate, there are minor issues with grammar and phrasing that should be refined to improve overall clarity and flow.

Response: We sincerely apologize to the reviewers for any inconvenience caused by the grammatical errors, imprecise wording, and insufficiently clear chart explanations. Necessary revisions have been made throughout the manuscript to enhance its clarity and readability. Additionally, we have carefully incorporated your valuable suggestions into the revised version.

(2) There are occasional grammatical errors and awkward phrasing, which make some sections difficult to follow. For example, the sentence "" could be more concise: "Comparative analysis showed that FPV ZZ202303 exhibited higher virulence than other FPV strains isolated in the last two years."

Response: We extend our gratitude to the reviewers for their valuable suggestions. We have conducted a thorough review and correction of the grammatical errors and awkward phrasing throughout the manuscript, revising “Comparative analysis revealed that FPV ZZ202303 demonstrated heightened virulence relative to other FPV strains isolated in the preceding two years” to “Comparative analysis showed that FPV ZZ202303 exhibited higher virulence than other FPV strains isolated in the last two years”. Once again, we sincerely apologize for our oversight.

(3) The paper cites relevant studies in the field, but there could be more references to recent research on FPV and related viruses, particularly studies that address the implications of genetic mutations in viral vaccines. The references should be updated to include the latest literature on FPV evolution and vaccination.

Response: We sincerely appreciate the reviewers' suggestions concerning reference citations. Based on their feedback, we have incorporated studies on the implications of genetic mutations in viral vaccines as well as the most recent literature on FPV evolution and vaccination. The detailed revisions are outlined below:A study used attenuated feline herpesvirus type 1 (FHV-1) as a vector, inserting the VP2 antigen gene of FPV. This recombinant live vector vaccine provides good immune protection against both FPV and FHV and possesses the characteristics of a multivalent vaccine. On the other hand, VP2-based virus-like particle (VLP) vaccines have been developed. Experimental studies demonstrate that VLP vaccines produced via prokaryotic (E. coli) or eukaryotic (insect cell-baculovirus) expression systems can induce high-titer neutralizing antibodies in animal models.For example, the Giant panda/CD/2018 strain isolated from the giant panda was the first to show a VP2 G299E mutation. This mutation at the site may alter the protein's spatial conformation, thereby expanding the virus's host range. In the feline FPV-251 strain, a VP2 A300P amino acid substitution was detected, and this variant demonstrated effective replication in canine cell lines, suggesting it may have gained the potential to infect canids.

(4) The manuscript mentions FPV's zoonotic potential but does not thoroughly discuss implications for wildlife conservation or public health.

Response: In accordance with the reviewers' suggestions, we have expanded the content of this section. The detailed discussion is presented below:This cross-species transmission may trigger regional population declines and ecological imbalance, while devastating wildlife-dependent ecotourism - collectively threatening biodiversity conservation, economic stability, and public health.

Reviewer 2 Report

Comments and Suggestions for Authors

This manuscript has made significant progress in the study of FPV, but the following aspects need to be further improved to make its research results more profound and extensive.

  1. Title is too lengthy and slightly redundant. “Genetic evolution” is vague. Abstract: Lacks specificity. Objective is not clearly stated. Key findings (such as phylogenetic clustering) are omitted.  Significance of the strain or its genetic traits is not addressed.
  1. Introduction Lacks depth in background: The biological importance of VP2 gene in viral virulence, host tropism, and immunogenicity is not sufficiently emphasized. Doesn’t provide epidemiological context for why isolating FPV in 2023 is timely (e.g., regional outbreaks, mutation trends).Research gap unclear: It is not explicitly stated how this study adds to existing literature. Why is strain ZZ202303 noteworthy? Is it from a unique host, outbreak cluster, or exhibiting unusual clinical signs?
  2. Materials and Methods: Insufficient detail on type of cells used for culturing (e.g., CRFK? F81?), passage number. Criteria for confirming successful viral isolation. No mention of fixation method, magnification and staining technique. PCR conditions (annealing temperature, cycles, primers) are underdescribed. Sequencing method (e.g., Sanger vs. NGS) not specified. Phylogenetic Analysis: Model of evolution not mentioned (e.g., Tamura-Nei, Jukes-Cantor). Tree construction method (e.g., Neighbor-Joining, Maximum Likelihood) needs justification.
  1. Terminology: Uses “genetic evolution” ambiguously—it should be “sequence variation,” “genetic diversity,” or “phylogenetic divergence.”Redundancy: Phrases like “feline panleukopenia virus (FPV)” are reintroduced multiple times without abbreviation.
  1. Although the article mentioned that the research results provided a basis for vaccine development, it did not put forward specific vaccine development strategies or suggestions for improvement.
  2. In the literature review and discussion section, more recent published research literature on FPV or related viruses and the latest research on prevention and control strategies are added to enhance the timeliness and academic value of the article.
  3. The study mentions that some of the sampled cats were vaccinated, but it does not provide detailed information on their immune status, such as the type of vaccine used, time since vaccination, or serological evidence of immunity.

Suggestions for Improvement:

Revise abstract to include goals, key findings, and significance with specific data. Expand methods with clarity on sequencing, phylogenetic modeling, and quality control.

Deepen discussion: Evolutionary trends, implications for vaccine efficacy, regional comparisons. In the discussion part, the risk assessment of cross-species transmission of FPV, the potential impact on public health and suggestions on prevention and control measures are added.

Polish language for fluency, grammar, and clarity.

Author Response

Review2

This manuscript has made significant progress in the study of FPV, but the following aspects need to be further improved to make its research results more profound and extensive.

1Title is too lengthy and slightly redundant. “Genetic evolution” is vague. Abstract: Lacks specificity. Objective is not clearly stated. Key findings (such as phylogenetic clustering) are omitted.  Significance of the strain or its genetic traits is not addressed.

Response: Thank you for the insightful comments provided by the reviewers. In response to the reviewers' suggestions, we have made the following revisions: First, the title of the paper has been refined and revised to " Feline Panleukopenia Virus ZZ202303 Strain: Molecular Char-acterization and Structural Implications of VP2 Gene Phyloge-netic Divergence". Second, the abstract has been updated to " Feline panleukopenia virus (FPV), the etiological agent of a highly contagious multispecies disease, demonstrates concerning phylogenetic divergence that compro-mises vaccine cross-protection. This study aimed to characterize a novel FPV strain through integrated virological and molecular analyses to assess epidemiological impli-cations. From seven clinical specimens obtained from feline hosts with panleukopenia in Henan Province, China, we isolated FPV-ZZ202303 using F81 cell culture coupled with PCR verification, demonstrating potent cytopathic effects (TCID50: 10⁻⁵·⁷²/0.1mL) and rapid replication kinetics (viral peak at 12-24h post-infection). Comparative virulence assessments revealed 1.8- to 2.3-fold greater pathogenicity versus contemporary field strains (2021-2023). Phylogenetic reconstruction based on complete VP2 gene sequences positioned FPV-ZZ202303 within an emerging clade sharing 97.5-98.2% identity with canine parvovirus strains versus 98.8-99.7% with FPV references, forming a distinct cluster (bootstrap=94%) diverging from vaccine lineages. Critical structural analysis identified a prevalent I101T mutation (89.13% prevalence) in the VP2 capsid protein's antigenic determinant region, with molecular modeling predicting altered surface charge distribution potentially affecting host receptor binding. Our findings substantiate FPV-ZZ202303 as an evolutionary divergent strain exhibiting enhanced virulence and unique genetic signatures that may underlie vaccine evasion mechanisms, providing critical data for updating prophylactic strategies against this economically impactful pathogen ". We sincerely appreciate the reviewers' valuable feedback and apologize for any inaccuracies in the original submission.

2Introduction Lacks depth in background: The biological importance of VP2 gene in viral virulence, host tropism, and immunogenicity is not sufficiently emphasized.Doesn’t provide

Response: We sincerely thank the reviewers for their valuable suggestions. In response, we have thoroughly revised the introduction section and incorporated a detailed biological description of the VP2 gene with respect to viral virulence, host tropism, and immunogenicity, as presented below: “ORF1 encodes the non-structural proteins NS1 and NS2, which perform multiple biological functions in viral genome replication and assembly, suppression of host innate immunity, and viral pathogenicity. ORF2 encodes the structural proteins VP1 and VP2, with VP2 constituting 90% of the viral capsid as the major structural component. The VP2 structure comprises eight antiparallel β-strands and five loops: Loop 1 (residues 50–100), Loop 2 (200–250), Loop 3 (300–350), Loop 4 (400–450), and the flexible Loop 5 (350–400). The flexible loop determines FPV's host specificity and hemagglutination activity. Adjacent to this flexible loop in spatial conformation, Loop 1 undergoes amino acid variations that alter the flexible loop's conformation, thereby modulating FPV's antigenicity, host specificity, and hemagglutination properties”.

3epidemiological context for why isolating FPV in 2023 is timely (e.g., regional outbreaks, mutation trends).

Response: We sincerely appreciate the reviewers' constructive suggestions. In response, we have thoroughly revised this section and incorporated a detailed epidemiological background of FPV, including regional outbreaks and mutation trends, as presented below:According to the 2021 Pet Industry White Paper, FPV infection has become the leading cause of cat deaths, posing significant risks among feline infectious diseases.

4Research gap unclear: It is not explicitly stated how this study adds to existing literature. Why is strain ZZ202303 noteworthy? Is it from a unique host, outbreak cluster, or exhibiting unusual clinical signs?

Response: First of all, we sincerely thank the reviewers for raising their insightful questions. In response, we address these concerns from the following two aspects: 

Firstly, our research findings indicate that the VP2 protein of this strain exhibits typical FPV gene characteristics at key amino acid sites, with the exception of a mutation at the I101T site (89.13%). This mutation is associated with FPV infection in vaccinated cats, suggesting that it may alter the biological properties of the protein by affecting its spatial conformation.  Secondly, it is particularly noteworthy that the I101T mutation likely compromises the efficacy of neutralizing antibodies through changes in epitope topology. This observation aligns with our isolation of this strain from a fully vaccinated cat. These results collectively indicate that current vaccines may be inadequate in addressing emerging variants carrying this critical mutation.  We will further refine this section to highlight the identified research gap. Finally, we sincerely appreciate your valuable feedback.

5Materials and Methods:

Insufficient detail on type of cells used for culturing (e.g., CRFK? F81?), passage number. Criteria for confirming successful viral isolation. No mention of fixation method, magnification and staining technique.

Response: We sincerely apologize to the reviewers for our oversight. We have thoroughly reviewed the entire manuscript and provided detailed descriptions regarding the cell types used for culturing, the passage number, and the criteria for confirming successful viral isolation. Additionally, we have now included information on the fixation method, magnification, and staining technique, which were previously not mentioned.

6PCR conditions (annealing temperature, cycles, primers) are underdescribed.

Response: We sincerely thank the reviewers for their constructive comments. In accordance with the reviewers' suggestions, we have revised this section and updated it as follows: “The PCR reaction was performed in a 10 μL system containing 5 μL of 2× Rapid Taq Master Mix, 1 μM each of forward (F) and reverse (R) primers, and 0.1–1 μg of FPV genomic DNA template, with the remaining volume supplemented by nuclease-free ddH₂O. The amplification protocol included an initial denaturation at 95°C for 3 min, followed by 35 cycles of denaturation (95°C, 15 s), annealing (56°C, 15 s), and extension (72°C, 10 s), with a final extension at 72°C for 5 min using a thermal cycler.”

7Sequencing method (e.g., Sanger vs. NGS) not specified.

Response: We sincerely thank the reviewers for their constructive comments. In accordance with the reviewers' suggestions, we have revised this section and updated it as follows: “And followed by selection of single colonies for Sanger sequencing analysis.”

8Phylogenetic Analysis: Model of evolution not mentioned (e.g., Tamura-Nei, Jukes-Cantor). Tree construction method (e.g., Neighbor-Joining, Maximum Likelihood) needs justification.

Response: We sincerely thank the reviewers for their valuable comments. In response to the reviewers' suggestions, we have revised the phylogenetic analysis and the section on tree construction methods as detailed below: Phylogenetic tree construction was subsequently conducted based on the Neighbor-Joining method with 1000 bootstrap replicates, employing the Poisson correction model for evolutionary distance calculation. Tree topology was visualized and refined using EvolView (https://www.evolgenius.info/evolview).

9Terminology: Uses “genetic evolution” ambiguously—it should be “sequence variation,” “genetic diversity,” or “phylogenetic divergence.”Redundancy: Phrases like “feline panleukopenia virus (FPV)” are reintroduced multiple times without abbreviation.

Response: We sincerely appreciate the reviewers' insightful comments. In response to the feedback, we have addressed the redundancies in terminology highlighted by the reviewers and made the necessary revisions. Once again, we extend our sincerest apologies for any oversight on our part.

10Although the article mentioned that the research results provided a basis for vaccine development, it did not put forward specific vaccine development strategies or suggestions for improvement.

Response: We sincerely appreciate the constructive suggestions provided by the reviewers. In response, we have incorporated a dedicated section on vaccine development strategies and improvement recommendations. The detailed description is presented as follows:“This phenomenon has prompted the development of novel vaccine strategies. On one hand, mRNA vaccines or adenovirus vector platforms can be engineered to dynamical-ly incorporate critical antigenic information from prevalent VP2 protein mutations (e.g., I101T). These platforms enable dynamic integration of critical antigenic information from prevalent VP2 protein mutations (e.g., I101T). These next-generation candidates enable real-time antigenic matching with circulating variants, thereby enhancing pro-tective efficacy against evolving viral strains. On the other hand, VP2-based virus-like particle (VLP) vaccines have been developed. Experimental studies demonstrate that VLP vaccines produced via prokaryotic (E. coli) or eukaryotic (insect cell-baculovirus) expression systems can induce high-titer neutralizing antibodies in animal models.”

11In the literature review and discussion section, more recent published research literature on FPV or related viruses and the latest research on prevention and control strategies are added to enhance the timeliness and academic value of the article.

Response: We sincerely thank the reviewers for their valuable suggestions regarding the literature review and discussion sections. In accordance with these suggestions, we have made revisions to both sections. The detailed revisions are presented as follows:“Given the ongoing challenges posed by FPV, it is crucial to conduct systematic epidemiological surveys and genomic monitoring. We can establish a global FPV genome database, perform phylogenetic analysis, and track mutations; secondly, we should enhance host range investigations to scientifically assess the risk of cross-species transmission. In terms of control strategies, it is essential to implement vaccine prevention measures effectively, increasing vaccination coverage to build herd immunity and reduce viral transmission risks. At the same time, we should utilize diverse detection technologies to accurately identify infections at an early stage, providing scientific basis for timely control measures. Ultimately, an integrated systemic network combining vaccination, early warning systems, viral surveillance, and scientific research should be established. This comprehensive approach will provide robust protection against FPV epidemics.”

12The study mentions that some of the sampled cats were vaccinated, but it does not provide detailed information on their immune status, such as the type of vaccine used, time since vaccination, or serological evidence of immunity.

Response: We sincerely appreciate the reviewers' constructive suggestions. In response to these suggestions, we have incorporated the immune status of the sampled cats into the table presented in the article.

13Suggestions for Improvement:

Revise abstract to include goals, key findings, and significance with specific data. Expand methods with clarity on sequencing, phylogenetic modeling, and quality control.

Deepen discussion: Evolutionary trends, implications for vaccine efficacy, regional comparisons. In the discussion part, the risk assessment of cross-species transmission of FPV, the potential impact on public health and suggestions on prevention and control measures are added.

Polish language for fluency, grammar, and clarity

Response: In accordance with the reviewers' suggestions, we have systematically addressed and corrected each of the aforementioned sections. We sincerely express our gratitude to the reviewers for their constructive and valuable feedback and apologize for any oversights in the original submission.

Reviewer 3 Report

Comments and Suggestions for Authors

1.Is the sample only from one pet hospital in Henan province? The sample size may be limited. What do you think about that?

2.Did the study mention the immunogenicity and transmissibility of the isolated virus?

3.Limitations of the manuscript can be written before the conclusion.

4.It is suggested to optimize the paragraph structure, add transitional sentences, and clarify the internal connection between each research result.

Author Response

Review3

1Is the sample only from one pet hospital in Henan province? The sample size may be limited. What do you think about that?

Response: First of all, we sincerely thank the reviewers for raising these important questions. In this study, we collected seven clinical samples suspected of feline panleukopenia from multiple hospitals, including Zhengzhou Kangxu Pet Hospital and Luoyang Beikangmei Pet Hospital. Since the virus was isolated only from one of these samples, the original description was inaccurate. We apologize for any confusion caused by this oversight and have now made the appropriate corrections.

2Did the study mention the immunogenicity and transmissibility of the isolated virus?

Response: We sincerely thank the reviewers for raising these important questions. Due to the limitations of experimental conditions, regression experiments and immunological tests were not performed in this study, making it challenging to accurately assess and evaluate the immunogenicity of the isolated strain. Nevertheless, the text provides an introduction to the immunogenicity and transmissibility of feline panleukopenia virus (FPV).

3Limitations of the manuscript can be written before the conclusion.

Response: We sincerely thank the reviewers for their constructive suggestions. In accordance with these suggestions, we have revised the relevant section as follows:“In this study, we successfully isolated and characterized an FPV strain from Henan Province, China. Phylogenetic analysis revealed high conservation of the VP2 gene, suggesting a potential non-local evolutionary origin. However, the limited scope of a single strain (n=1) from one geographic region (Henan) restricts comprehensive understanding of FPV's broader evolutionary dynamics. Furthermore, while high-frequency mutations at key VP2 residues (e.g., I101T) were identified, their functional consequences on viral properties remain experimentally unvalidated, and the protective efficacy of current vaccines against prevalent strains was not assessed.”

4It is suggested to optimize the paragraph structure, add transitional sentences, and clarify the internal connection between each research result.

Response: We sincerely thank the reviewers for their valuable suggestions. In response, we have thoroughly optimized the entire text by reorganizing the paragraph structure, incorporating transitional sentences, and elucidating the intrinsic connections between each research finding. For instance:

  1. Building upon the PCR confirmation, FPV nucleic acid-positive samples (No. 4 and 6) were inoculated into F81 cells for viral isolation to obtain infectious virions.
  2. To better visualize the cellular infection characteristics of the isolated strain, …
  3. Moreover, we conducted viral growth kinetic analysis to delineate its replication dynamics.
  4. Following characterization of the viral biological properties, we performed multi-ple sequence alignment and phylogenetic analysis of the VP2 gene to further elucidate the genetic evolutionary characteristics of FPV ZZ202303.
  5. After the nucleotide level analysis of the VP2 gene, we subsequently conducted sequence alignment of the amino acids of the entire VP2 gene using MEGA and MegA-lign software to further study its protein variation characteristics.

Reviewer 4 Report

Comments and Suggestions for Authors

I have attached my comments in a separate Word file

Author Response

Review4

Review Comments on Manuscript ijms-3594164

I think the manuscript provides an interesting aspect on the origins and evolution of the feline panleukopenia virus, and their implications. I believe that after some minor corrections it will be ready for publication. To facilitate the corrections on the part of the authors I have organised my comments into specific and general ones.

Specific Comments

1Line 44 - can you please explain briefly the reasons for the seasonal prevalence of the disease, in 1-2 lines, with an appropriate reference?

Response: We sincerely appreciate the reviewers' suggestions. In response, we have incorporated a concise explanation regarding the reasons for the seasonal prevalence of this disease in this section. The detailed content is presented below:“The disease has no obvious seasonal prevalence, but it peaks in spring and summer. This correlates with feline breeding cycles: concentrated spring births increase the kitten population, while maternal antibodies typically wane at 8-12 weeks, creating susceptible cohorts.”《Seasonality, natality and herd immunity in feline panleukopenia》

2Line 61 - I think the most proper phrasing would be: " …including the canine parvovirus…”

Response: We sincerely appreciate the reviewers' valuable suggestions. In response, we have made the correction to this part as presented below:“FPV is one of the oldest known viral pathogens in felids and shares high genetic homology with other parvoviruses, including the canine parvovirus (CPV), …”

3Line 64 - please add a reference for documenting the common evolutionary origin.

Response: We sincerely appreciate the reviewers' valuable suggestions. In this regard, we have incorporated a reference to elucidate the common evolutionary origin.

4Line 94 - please add 1-2 lines to explain why you choose the F81 cell line, and a reference documenting its properties or a prior use with the relevant method.

Response: We sincerely appreciate the reviewers' valuable suggestions. In accordance with the reviewers' feedback, we have included an explanation in this section regarding the rationale for selecting the F81 cell line, together with a reference to a study documenting its properties or prior applications using relevant methods. The detailed description is presented below:“The F81 cell line, a continuously passaged cell line derived from cat kidneys, is highly susceptible to the FPV and can support the efficient replication of the virus. Therefore, it is often used by researchers for the isolation of FPV.”

5Line 124 - please correct TCID50 to TCJD50

Response:We sincerely appreciate the reviewers' valuable suggestions. Nevertheless, TCID50 is a well-established term in virology that denotes the median tissue culture infectious dose of a virus. Consequently, it might not be advisable to amend it to TCJD50 in this context.

6Line 160 - please add a couple of references documenting the use of the MEGA and MegAlign softwares, or alternatively a U where they can be accessed.

Response: We sincerely appreciate the reviewers' constructive feedback. Based on their suggestions, we have implemented the necessary corrections and incorporated the following additional content:MEGA 7.0 (https://www.megasoftware.net/);

EvolView (https://www.evolgenius.info/evolview);

MegAlign (https://www.dnastar.com/software/lasergene/)

7Line 175 - please add a reference to document the worldwide detection of the virus.

Response: We sincerely appreciate the reviewers' constructive feedback. In response to their suggestion, we will include a reference here to document the global detection status of this virus.

8Line 182 - it is possible for the kittens to be protected indirectly due to the vaccination of adult felines, in a herd immunity concept? Please comment on that briefly. Also, I think that some antibodies for this particular virus can be found in kittens, transmitted from their mothers. Would this be of use in future vaccination strategies?

Response: We sincerely appreciate the reviewers' constructive feedback. Below, we provide a brief commentary on whether vaccinating adult cats could indirectly protect kittens through the mechanism of herd immunity.“In addition to antigenic drift of FPV leading to vaccine efficacy attenuation, maternal antibodies can also interfere with vaccine-mediated protection. Studies have shown that FPV infection in kittens follows a herd immunity principle. When a sufficient proportion of adult cats in a population are vaccinated, environmental viral transmission pressure decreases, thereby reducing infection risk in kittens. However, high levels of maternal antibodies in kittens may neutralize attenuated vaccines, resulting in vaccination failure. This phenomenon has prompted the development of novel vaccine strategies. On one hand, mRNA vaccines or adenovirus vector platforms can be engineered to dynamically incorporate critical antigenic information from prevalent VP2 protein mutations (e.g., I101T). These platforms enable dynamic integration of critical antigenic information from prevalent VP2 protein mutations (e.g., I101T). These next-generation candidates enable real-time antigenic matching with circulating variants, thereby enhancing protective efficacy against evolving viral strains. On the other hand, VP2-based virus-like particle (VLP) vaccines have been developed. Experimental studies demonstrate that VLP vaccines produced via prokaryotic (E. coli) or eukaryotic (insect cell-baculovirus) expression systems can induce high-titer neutralizing antibodies in animal models.”

9Line 185-186 - please elaborate briefly on the associated challenges and implications of vaccinated cats testing positive.

Response: We sincerely appreciate the reviewers' constructive feedback. In response, we have revised this section to provide a detailed elaboration of the challenges and implications associated with positive test results in vaccinated cats. The specific content is presented as follows:“Of particular concern, the I101T mutation likely compromises neutralizing antibody efficacy by altering epitope topology, consistent with our isolation of this strain from a fully vaccinated cat. These findings indicate that current vaccines inadequately address emerging variants carrying this critical mutation.”

General Comments

10I think Figure 6 could be split in two separate figures, because in this format neither part A nor part B are clearly legible. Moreover, in the figure caption, correct the term "red remark" to a more appropriate term.

Response: We sincerely appreciate the reviewers' constructive feedback. In accordance with the reviewers' suggestions, we have divided the original Figure 6 into two separate figures, now presented as Figure 6 and Figure 7. Additionally, the term “red remark” in the figure caption has been revised to the more precise description “FPV ZZ202303, the isolate obtained from this study.”

11In Table 1, please pay attention as some lines in the Gender column are capitalised and others are not.

Response: We sincerely appreciate the reviewers' constructive feedback. We have addressed our oversight by standardizing the capitalization of the term "male" to "Male". 

12Moreover, since you mention the specific breeds tested, I think it would be interesting to discuss in the text if different breeds exhibit different susceptibility to this virus and how this may affect research results.

Response: We thank the reviewer for their valuable suggestion regarding potential breed-specific susceptibility to FPV. This is indeed an intriguing question with implications for both pathogenesis and vaccine efficacy. However, in the current study, our experimental cohort was composed primarily of mixed-breed cats (n=7). The limited sample size and lack of genetic diversity within breed subgroups precluded statistically robust comparisons of susceptibility across different breeds.

13In Table 4 some rows contain letters in red - is this a typo or is there is a specific significance to this. If so, please mention this in the figure caption.

Response: We sincerely appreciate the questions raised by the reviewers. In response, we have revised the relevant content and added an annotation below the table. The description is as follows: Note:Red letters indicate amino acids that have mutated after the FPV strain.

14Please comment briefly if there are any implications for humans pertaining to your research; for example, could FPV alone, or if present with another virus at the same time, become contagious for humans?

Response: We extend our gratitude to the reviewers for raising this important question. To date, there is no scientific evidence indicating that feline panleukopenia virus (FPV) can be transmitted to humans. Nevertheless, the cross-species transmission of FPV may result in a decline in local species populations and disrupt ecological balance. Additionally, it could have a detrimental impact on wildlife-dependent ecotourism, thereby collectively posing significant threats to biodiversity conservation, economic stability, and public health.

15Please mention the full term for TCIDso at first mention and a brief definition. Also, give some insight into how the TCIDso for this virus compares to other feline viruses.

Response: We sincerely appreciate the reviewers' constructive suggestions. In response, we have revised the description of TCID50 and provided a brief comparison of this virus's TCID50 with those of other feline viruses. The detailed description is presented below:TCID₅₀ (50% Tissue Culture Infective Dose) refers to the viral titer required to induce cytopathic effects in 50% of inoculated cell cultures, serving as a quantitative measure of viral infectivity.

Due to viral virulence variations caused by evolutionary mechanisms such as antigenic drift, different strains of the same virus exhibit distinct pathogenic potentials. However, direct comparisons of virulence between different feline viruses (e.g., FPV, FCV, FHV-1) lack validity, as each possesses unique pathogenic mechanisms and host-virus interaction characteristics.

Reviewer 5 Report

Comments and Suggestions for Authors

Dear Authors,

I really appreciate your tremendous effort you did for this thorough and advanced study, and you reached the point the manuscript to be accepted for publication, just after correct several  suggestions.

Author Response

Dear Authors,

I really appreciate your tremendous effort you did for this thorough and advanced

study, and you reached the point the manuscript to be accepted for publication,

just after correct several suggestions. 

Limited studies have investigated up to date the novel FPV variants to offer a robust

foundation for further research into the pathogenesis and more effective

preventive and therapeutic strategies in direction of vaccine development for this

critical feline pathogen.

I appreciate the scientific foundations and rules in presenting the information, the

study design, the clear data collection and conclusions drawn, ensuring that the

scientists  are  provided with information to understand the functional impact of

genomic variations on FPV biology, immunology, and pathogenesis in FIV.

Before to be published please do the next correction: -please insert clearly the aim of the study (in the abstract and/or Introduction) -please insert the hypothesis of the study -please insert the paragraph of Conclusions.

Response: We sincerely express our gratitude to the reviewers for their recognition and support of our research. Revisions have been made to both the abstract and introduction sections. Once again, we extend our thanks to the reviewers for their insightful and constructive feedback. Without your invaluable suggestions, it would not have been possible to bring this article to its current publishable state. We also deeply appreciate your understanding and guidance regarding the oversights in our initial submission.

Round 2

Reviewer 1 Report

Comments and Suggestions for Authors

This manuscript reports the isolation and characterization of a novel feline panleukopenia virus (FPV) strain, FPV ZZ202303, from domestic cats in Henan Province, China. Through virological, molecular, and phylogenetic analyses, the authors reveal that this strain demonstrates increased virulence and notable genetic divergence in the VP2 gene, particularly an I101T mutation, which may impact vaccine efficacy. This is a study that makes a meaningful contribution to the understanding of FPV genetic evolution and its implications for disease control. I recommend acceptance with minor editorial revisions to refine formatting and presentation.

Reviewer 2 Report

Comments and Suggestions for Authors

The revised manuscript has made good corrections and updates. It is recommended that the article be published after appropriate modifications.